# Semiochemicals produced by fungal bark beetle symbiont *Endoconidiophora rufipennis* and the discovery of an anti-attractant for *Ips typographus*

**Matilda Lindmark[1], Suresh Ganji[2], Erika A. Wallin[3], Fredrik Schlyter[4,5], C. Rikard Unelius[2]***

**1** Biology, Department of Natural Sciences, Mid Sweden University, Sundsvall, Sweden, **2** Faculty of Health and Life Sciences, Linnaeus University, Kalmar, Sweden, **3** Eco-Chemistry, Department of Chemical Engineering, Mid Sweden University, Sundsvall, Sweden, **4** Czech University of Life Sciences Prague, Faculty of Forestry and Wood Sciences, Excellent Team for Mitigation (ETM), Suchdol, Czech Republic, **5** Chemical Ecology, Department of Plant Protection Biology, Swedish University of Agricultural Sciences, Alnarp, Sweden

* rikard.unelius@lnu.se

**Data Availability Statement:** All relevant data are within the paper and its Supporting Information files.

## Abstract

Bark beetles vector symbiotic fungal species into their host trees during mass attacks. The symbiotic relationship with blue stain fungi of the Ascomycetes, including genera of *Endoconidiophora* (syn. = *Ceratocystis*), promotes successful establishment whereby the microbes help to overcome the host trees' defence and degrade toxic resins. This is the first study to evaluate both the volatile emissions from an insect-associated blue stain fungus over time and the insect response in a field trapping experiment. Volatile emissions from isolates of *Endoconidiophora rufipennis* (ER) were collected by solid-phase microextraction (SPME) and analysed by gas chromatography—mass spectroscopy (GC-MS) over a period of 30 days. This virulent North American fungus is closely related to *E. polonica*, a symbiotic fungus known from Eurasian spruce bark beetle *Ips typographus*. Nine volatiles were emitted by ER in substantial amounts: isoamyl acetate, sulcatone, 2-phenethyl acetate, geranyl acetone, geranyl acetate, citronellyl acetate, (*R*)- and (*S*)-sulcatol, and (*R*)-sulcatol acetate. A late peaking compound was geranyl acetone. In the field trapping experiment, three of the fungal volatiles (geranyl acetone, 2-phenethyl acetate and sulcatone) were tested in combination with a synthetic aggregation pheromone for *I. typographus*. Traps with geranyl acetone attracted lower numbers of *I. typographus* compared to traps with 2-phenethyl acetate, sulcatone or the pheromone alone as a control. The results showed that geranyl acetone acts as an anti-attractant and may act naturally on *I. typographus* as a cue from an associated fungus to signal an overexploited host.

## Introduction

The Eurasian spruce bark beetle *Ips typographus* (L.) is one of the most serious pests of Norway spruce across the Palearctic. Generally, conifer bark beetles attack old, unhealthy trees

**Funding:** This work was supported by two grants from The Carl Trygger Foundation, (CRU and SG), and Linnaeus University, Kalmar, Sweden. This research was also financially supported by the Regional Development Fund for Västernorrland (20201370), Jämtland/Härjedalen (RUN/94/2017 ) and Västernorrland (17RS484) county and the Swedish Agency for Economic and Regional Growth (EAW and ML). FS was supported by project EXTEMIT-K CZ.02.1.01/0.0/0.0/15_003/0000433 financed by OP RDE at the Czech University of Life Sciences, Prague. There was no additional external funding received for this study. The funders had no role in study design, data collection and analysis, decision to publish, or preparation of the manuscript.

**Competing interests:** The authors have declared that no competing interests exist.

weakened by disease, drought, stress, or physical damage. Healthy trees can defend themselves by producing resin, which contains insecticidal and fungicidal compounds that can stop or arrest attacking insects [1–3]. However, during outbreaks, beetle populations may reach threshold densities where intensive mass attacks can kill apparently healthy trees [4, 5]. The ongoing climate change, with rising temperatures and more drought events, may trigger even more severe and frequent outbreaks [6]. Millions of hectares of commercial [7, 8] and natural [9] spruce forests have been killed by *I. typographus* in Europe. Blue stain fungi are commonly associated with bark beetles, including *I. typographus*, and may help the beetles to overcome the defences of the conifer host tree [10–12]. Bark beetles in Europe transport blue stain fungi from the genera *Endoconidiophora* (syn. = *Ceratocystis*), *Ophiostoma* and *Grosmannia* [10, 13].

Across the Northern Hemisphere, there is growing interest in volatiles released by blue stain fungi associated with noxious bark beetle species. Volatiles from symbiotic fungi, including *Endoconidiophora* spp, and yeasts present in the beetle alimentary canal can act as anti-aggregation cues by the oxygenation of compounds (e.g. verbenols to verbenone) [14–17]. In Europe, a symbiosis is well described between *I. typographus* and *E. polonica*. Extensive research has been performed regarding the ecology and possible convergent evolution of inter-specific semiochemical volatiles from trees, beetles, and fungi. Interspecific semiochemicals are volatiles released from one individual which affects the behaviours of an individual of another species. It is known that *I. typographus* responds to volatiles emitted from *E. polonica* at short range (cm) and that *E. polonica* helps the insects to colonise new trees [13, 18, 19].

The chemicals emitted by bark beetle associated blue stain fungi are often biosynthesised via isoprenic pathways, resulting in methylbutene derivatives (e.g., methylbutenols), oxygenated monoterpenes (e.g., verbenone, verbenol) and chain-shortened isoprenoids (e.g., sulcatone, geranyl acetone). With growing knowledge of the biosynthesis of fungi-emitted volatiles, it has been realised that the fungi and bark beetles both produce the active semiochemicals (including non-isoprenoids, such as *trans*-conophthorin) [18, 20]. The possibility of identifying new bark beetle semiochemicals (attractants or anti-attractants) was the main motivation for the present study of bark beetle associated fungal volatiles in the field. An attractant is a volatile which can be proven to attract individuals of a species, while an anti-attractant is an active compound which inhibits attraction of the same species. Such knowledge not only helps to understand their role in natural systems' functioning but may also enable their biotechnological production by yeast and fungi *in vitro* or *in vivo* [21].

To better understand the symbiotic relationship between *I. typographus* and *Endoconidiophora* and broaden the arsenal of potential anti-aggregation cues, we evaluated volatiles from the Ascomycotic fungus *E. rufipennis* (ER). ER is closely related to *E. polonica* and has been found associated with the North American spruce beetle *Dendroctonus rufipennis* [22, 23]. A potential symbiotic relationship between ER and *I. typographus* would parallel the confirmed relationship between *E. polonica* and *I. typographus*. It is important to bear in mind that ongoing climate change is redrawing the geographical ranges of species and allowing the establishment of invasive species [24, 25]. A future symbiotic relationship between *I. typographus* and ER might be possible due to the expanding geographical range of *I. typographus* from the Palearctic to North America. *I. typographus* is frequently detected at U.S. ports [26, 27]. A recent study showed that *I. typographus* was attracted to ER and able to colonise North American tree species such as black spruce (*P. mariana*) and white spruce (*P. glauca*) in a laboratory bioassay [24]. New associations could probably be formed and maintained over time, which in turn could enable the successful establishment of *I. typographus* in North America. Therefore, it is wise to prepare for new geographical establishments and better understand the potential of ER´s biological activity for *I. typographus*.

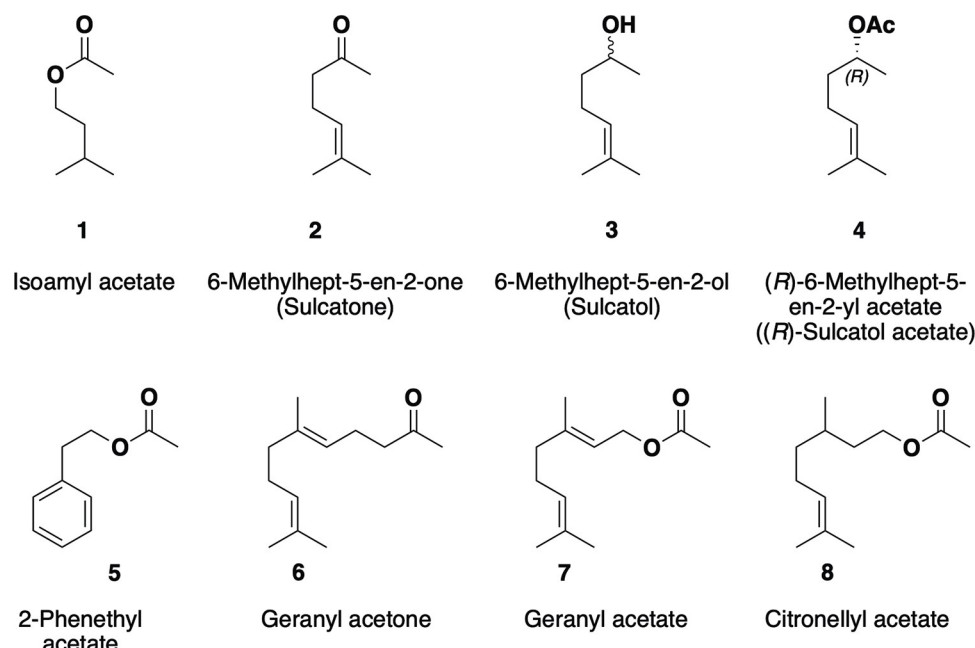

**Fig 1. Major volatiles identified from *Endoconidiophora rufipennis*.**

The present study investigated volatiles produced over time by the Nearctic blue stain fungus ER *in vitro* and evaluated the potential attractant or anti-attractant activity of the volatiles for *I. typographus*. Such semiochemicals could be used as part of an arsenal of tools to prevent the insects from establishing a viable population in North America and overcome epidemic outbreaks within the existing range.

So far, most research on the insect-fungus pair *I. typographus- Endoconidiophora* has focused on laboratory experiments assessing antennal activity or utilising short-range bioassays of identified compounds [18, 24, 28, 29]. To further understand the intricate relationship between bark beetles and symbiotic fungi, there is a need for field trapping experiments to evaluate the ecological relevance and potential of using compounds identified from fungal volatiles as attractants or anti-attractants over a longer distance. This is the first study to compare chemical analysis in the laboratory of volatiles from identified fungi in the bark beetle holobiont to insect behaviour in a field experiment. This paper presents these two aspects in sequence.

1. Analysis of volatiles emitted over time by *Endoconidiophora rufipennis*, an Ascomycotic fungus associated with the North American spruce beetle *Dendroctonus rufipennis* (Figs 1–4)

2. Evaluation by field trapping three selected physiologically active volatiles together with the aggregation pheromone of *Ips typographus* regarding their attractant or anti-attractant effects (Fig 5)

## Method and materials

### Fungal cultures

The fungal species *E. rufipennis* (ER), 1992-633/280/7, was collected in North America and supplied by Prof. Krokene, Norwegian Institute of Bioeconomy Research, Ås, Norway (more information in supporting information; **S1 Table**).

A 250 mL oven-dried glass bottle filled with 100 mL distilled water, 1.2 g Bacto Agar (Nordic Biolabs) and 840 mg LP 0039 Oxoid malt extract was covered with aluminium foil and autoclaved at 120˚C for 2 hours. After autoclaving, the bottle was allowed to cool to 60–70˚C and then placed in a laminar flow hood. The contents of the bottle were divided into six oven-dried 100 mL E-flasks (10 mL in each), which were laid almost horizontally in the flow hood. After the agar medium had solidified, a small piece of the stock fungus was inoculated in the centre of it using a sterile loop. Three E-flasks were inoculated with ER and the other three were marked as negative controls. The flasks were subsequently sealed with aluminium foil.

## SPME analysis

Solid-phase microextraction (SPME) was conducted using a 50/30 μm SPME fibre coated with divinylbenzene/carboxen/-polydimethylsiloxane (DVB/CAR/PDMS, Stableflex, Supelco) with needle size of 24 Ga. The fibre was preconditioned in the GC inlet prior to volatile collection. The extraction procedure was carried out in the gas phase of the culture flask without the fibre touching the culture by the headspace SPME method at ambient temperature. Volatile emissions of the fungus and controls (described above) were measured from day 2 to day 30 at intervals of 3–4 days. On the test days, the aluminium foil lids were punctured with the SPME holder, the fibre was ejected and the headspace in the flasks extracted for 45 min. After extraction, the hole in the aluminium foil was closed with a piece of tape and volatiles absorbed by SPME were desorbed in the GC-MS inlet at 220˚C for 5 minutes (these 5 minutes also served as conditioning of the SPME fibre). The SPME samplings of the fungus and malt agar controls were executed in triplicate.

## GC-MS analysis

The GC-MS instrument was a 6890 GC and 5973 mass selective detector (Hewlett Packard, Palo Alto, CA, USA). Helium was used as the carrier gas, and a polar enantioselective capillary column was used (Cyclosil B, 30 m × 0.25 μm, ID 0.25 mm, J&W Scientific, USA). Mass spectra were obtained by electron impact ionization (70 eV). The transfer line temperature was 280˚C, and the temperature program was as follows: initial temperature 40˚C (held for 3 min), raised to 150˚C at 3˚C min$^{-1}$ and then to 250˚C at 15˚C min$^{-1}$ (held for 10 min). The sample was desorbed in the splitless mode. The total run time was 56.3 min.

The fungal volatiles were tentatively identified using Wiley and NIST MS libraries. Subsequently their identities were confirmed by the analysis of reference compounds.

The amounts reported are relative to the total emission of volatiles in a sample (Table 1), except in Figs 2 and 4, where the relative amounts of specific semiochemicals are visualised.

**Table 1. Relative amounts (%) of total volatiles emitted by ER over time by SPME analysis.**

| Compound | Compound No. | 2nd day of analysis | Max % (at sampling day) | At end of analysis | Negative controls |
|---|---|---|---|---|---|
| Isoamyl acetate | **1** | 35 | 54 (6) | 0 | 0 |
| Sulcatone | **2** | 61 | 95 (18) | 94 | 0 |
| Sulcatol | **3** | 0 | 2 (25) | 1 | 0 |
| (*R*)-Sulcatol acetate | **4** | 0 | 0.2 (6) | 0 | 0 |
| 2-Phenethyl acetate | **5** | 1 | 1 (2) | 0 | 0 |
| Geranyl acetone | **6** | 0.8 | 5 (30) | 5 | 0 |
| Geranyl acetate | **7** | 0 | 0.6 (12) | 0 | 0 |
| Citronellyl acetate | **8** | 0.6 | 0.9 (4) | 0 | 0 |

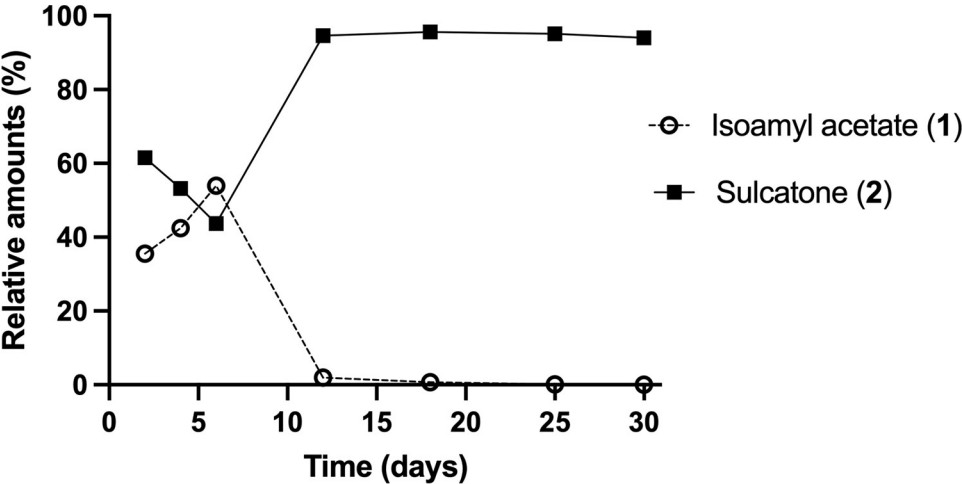

**Fig 2. Time course of relative amounts of isoamyl acetate (1) and sulcatone (2) and produced in an ER culture over time.**

## Chemicals

Starting materials and reactants were obtained from commercial sources (Aldrich, Germany or Chemtronica, Sweden) and used without further purification unless stated otherwise. The chemicals bought from Aldrich were CALB enzyme, sulcatone **2** (6-methyl-5-hepten-2-one, 98% purity), geranyl acetone **6** ((*E*)-6,10-dimethylundeca-5,9-dien-2-one, 96% purity), while the chemicals obtained from Chemtronica were: methanol (99% purity), sodium borohydride (96% purity), vinyl acetate (99% purity), citronellol (3,7-dimethyloct-6-en-1-ol 95% purity), isoamyl acetate **1** (3-methylbut-1-yl acetate, 98% purity), 2-phenethyl acetate **5** (2-phenylethyl acetate, 98% purity), and geranyl acetate **7** ((*E*)-3,7-dimethylocta-2,6-dien-1-yl acetate, 97% purity).

For confirmation of sulcatol isomers **3** (6-methylhept-5-en-2-ol) and sulcatol acetate isomers **4** (6-methylhept-5-en-2-yl acetate), syntheses were performed. The methods are described in in supporting information **S1 File**. The reference chemical citronellyl acetate **8** (3,7-dimethyloct-6-en-1-yl acetate) was synthesised by standard acetylation of citronellol with acetic anhydride.

## Field test

**Dispensers.** Three chemicals were selected for testing in the field, based on published reports of their activity in bark beetles (see Discussion for detailed rationale). The chemicals were tested in the field trapping experiment (together with a commercial aggregation pheromone for *I. typographus*): geranyl acetone (**6**), 2-phenethyl acetate (**5**) and sulcatone (**2**) (Table 2). All traps were baited with one bag of Ipslure® pheromone dispenser (Ipslure®, Kemikonsult, Norway) and one dispenser with the synthetic fungal compounds or an empty dispenser acting as a control. In total, 4 different treatments were tested.

All dispensers comprised 4 × 5 cm polyethylene bags (80 μm, O. Möllerström AB, Gothenburg Sweden) with a surface area of 3 × 5 cm containing a 3.5 × 1.3 cm piece of felt (100% viscose, Ernst Textil, Östra Karup, Sweden). For each of the test fungal compounds, 1 mL of synthetic pure compound was distributed onto the felt in a dispenser bag before it was sealed using a Quick Seal (Quick Seal 200, O. Möllerström AB, Gothenburg Sweden). As a control, a clean and dry piece of felt was used. Before testing the dispensers in the field, their release rate was measured in triplicate under laboratory conditions for over a week. By gravimetry, the

**Table 2. Chemicals, release rates and dispensers used in field trapping experiments.**

| Fungal volatiles in field trapping experiment | Purity | Release rate (mg day$^{-1}$)[a] | Dispenser type[b] | Loading (ml) | Ecological relevance | Previously known origin |
|---|---|---|---|---|---|---|
| Geranyl acetone (6,10-dimethyl-undeca-5,9-dien-2-one) **6** | 96% | 10 | PE bags | 1 | Anti-attractant for *I. subelongatus* (Zhang et al. 2007) | Non-host volatile from angiosperm leaves |
| 2-Phenethyl acetate **5** | 98% | 20 | PE bags | 1 | General volatile, symbiosis with *I. typographus* (Kandasamy et al. 2016) | Fungi: *E. polonica*, *G. pencillata* and *G. europhioides* |
| Sulcatone (6-methyl-5-hepten-2-one) **2** | 98% | 60 | PE bags | 1 | Isoprene analogue of geranyl acetone (Francke et al. 1995) | Hindgut in various bark beetles |

[a] Tested under laboratory conditions at 22°C in triplicate over a week. Number shows mean release rate.

[b] $4 \times 5$ cm (80 μm) polyethylene (PE) bags with surface area of $3 \times 5$ cm containing a $3.5 \times 1.3$ cm piece of felt (100% viscose). For each dispenser, 1 mL of fungal compound was distributed onto the felt in the dispenser bag before it was sealed using a Quick Seal.

dispensers were found to evenly release approx. 20 mg/day 2-phenethyl acetate (**5**), 60 mg/day sulcatone (**2**) and 10 mg/day geranyl acetone (**6**) at 22°C (see supporting information **S1 Fig**). The dispensers were kept in the freezer (−20°C) until they were transported to the field sites.

**Trapping.** Lindgren funnel traps with 10 funnel units [30] were set up in fresh clear-cuts of spruce forests in areas with high populations of *I. typographus* in central Sweden. No killing agent was used. Three areas were chosen for the experiment (62° 31–32′ N 16° 37–39′ E) which all fulfilled the following criteria: i) Outbreak of *I. typographus* during previous summer; ii) Surrounding forest stands of minimum 85% spruce (basal area) and minimum 60 years old; iii) Clear-cut performed from November in the previous year to March in the year of the trial; iv) No soil scarification, no plantation.

The experimental design was randomised blocks with three blocks (three clear-cut areas), resulting in 12 traps in total. Each trap was baited with one bag of Ipslure® together with one of the synthetic fungal compounds (or with an empty dispenser as a control). In each block, four traps were placed in line > 30 m from a sun-exposed forest edge with 30 m between each trap. The initial trap positions were randomised.

The experiment was started in the middle of May and continued for 4 weeks, covering the main flight period of *I. typographus* in the area. The traps were emptied every week during the trapping period (4 collections per trap) and rotated after each collection to avoid position effects. Insects were collected in separate plastic bottles and stored in a freezer (−20°C). After the field trial, the catches were counted and all specimens that were not *I. typographus* excluded. As we were only interested in total catch, the caught *I. typographus* were not sex separated.

**Trapping statistics.** The trap catch was first analysed as number ($x$, raw catch data), log catch number ($\log(x+1)$) and relative catch ($r = x/\Sigma$ per rotation in a block) to find a suitable form for analytical statistics. The data were evaluated by factorial ANOVA (Analysis of Variance) of all combinations of treatment and block effects and their interaction (SPSS V28.0 UNIANOVA). Subsequently, a generalised linear model was used to analyse the significance of treatment effects (SPSS V28.0 GENLIN). Alternatively, the effect sizes of differences in the catch between the three treatments and the control (Ipslure® and empty dispenser containing felt) were evaluated. The α-value for statistical tests was set at 0.05.

## Results

### Volatiles produced by *Endoconidiophora rufipennis*

Nine volatiles were emitted in substantial amounts by ER, most of them known semiochemicals (Fig 1) [31]. A typical gas chromatogram of fungal emissions recorded on day 10 is

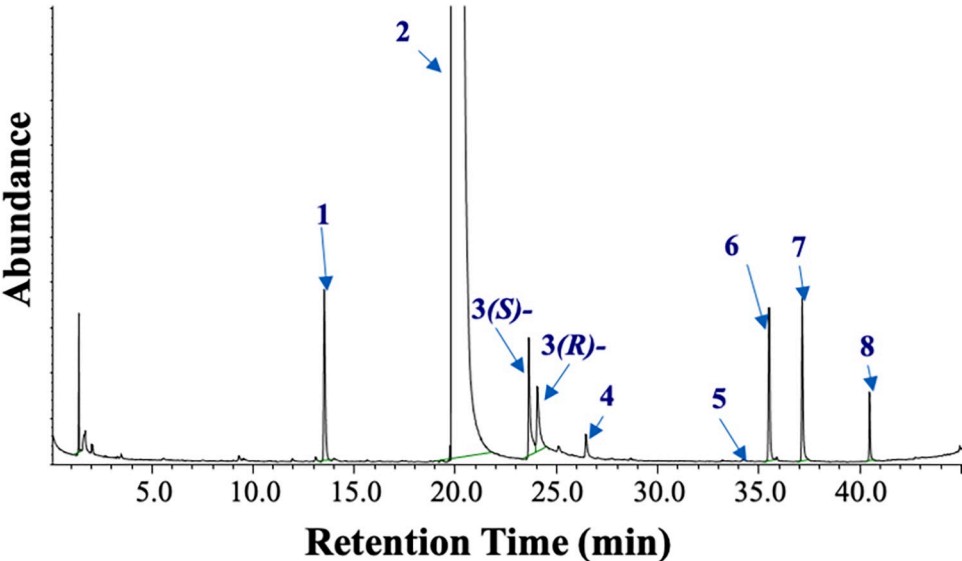

**Fig 3. Total ion chromatogram of volatiles collected from the headspace of ER cultivated on pure agar medium on day 10.** An enantioselective Cyclosil B column was used to separate stereoisomers. Peak information is given in Table 1.

depicted in Fig 3. A chromatogram of emissions from the control is provided in supporting information; **S1 File**.

ER produced different proportions of the compounds over time (Figs 2 and 4). Isoamyl acetate (**1**) was produced in high proportions in the agar medium between day 2 and 8, and then the production ceased completely (Table 1; Fig 2). In the malt agar controls, none of the compounds emitted by ER (Table 1) were detected. The peak abundance of each compound during analysis is presented in Table 1 and Figs 2 and 4. The amount of each compound varied with time and the peak abundance occurred at different times for the nine compounds.

Sulcatone (**2**, Figs 1–4) was emitted from ER and gradually increased from day 6 to the end of the analysis period (Fig 2). The relative amount of isoamyl acetate (**1**) was highest (> 55%) on day 6 (Fig 2). When the production of isoamyl acetate (**1**) ceased, sulcatone emission increased to 80–95% of total emitted volatiles for the rest of the period (18 days). On the final day of analysis,

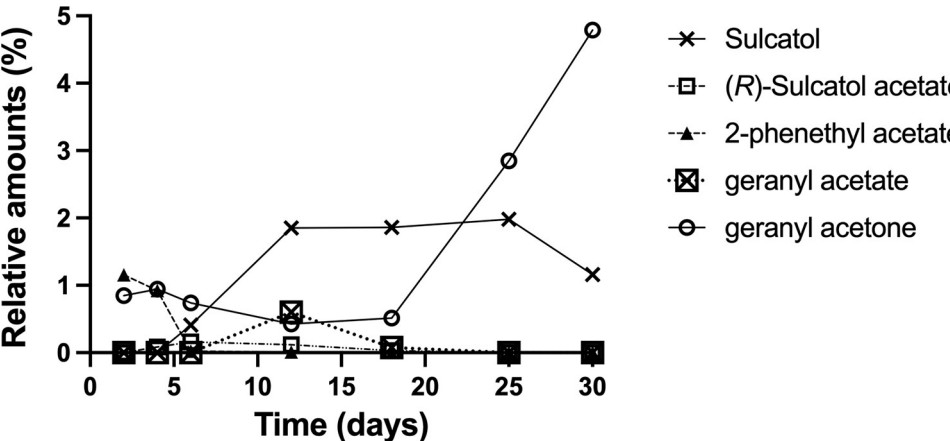

**Fig 4. Time dependence of relative amounts of compounds (3–7) produced in an ER culture (Citronellyl acetate (8) omitted for clarity, see Table 1 for data).**

sulcatone (**2**) represented > 91% of all emitted volatiles. We also found that ER emitted both stereoisomers of sulcatol ((**3**); Figs 1, 2). At the start of the analysis, there was no sulcatol (**3**) formed, but later a small amount gradually appeared in the culture. On the last day of analysis, the sulcatol enantiomers combined represented approximately 2% of the emissions.

The (*S/R*)-sulcatol (**3**) and (*R*)-sulcatol acetate (**4**) ratios produced in the agar medium varied during the analysis period. (*R*)-Sulcatol was initially produced in higher amounts than the (*S*)-isomer. In the middle of the analysis period, the ratio of isomers was 65:35 *S/R* (Fig 3), whereas at the end of the analysis period, it was 40:60 *S/R*. (*R*)-Sulcatol acetate (**4**) was produced with high enantioselectivity (Fig 3). The enantiomeric excess of (*R*)-sulcatol acetate (**4**) emitted was close to 90%. To confirm the absolute configuration of (**4**) emitted by the fungus, it was synthesised from sulcatol by enzymatic esterification using *Candida antarctica* lipase B and vinyl acetate. Racemic sulcatol was synthesised from 6-methyl-5-hepten-2-one (sulcatone, **2**, Aldrich, Germany) by using $NaBH_4$ in methanol (see supporting information; **S1 File**).

Geranyl acetone (**6**) increased to its maximum value (5%, Fig 4) on the last sampling day. ER also produced 2-phenethyl acetate (**5**) in small proportions (< 5%), and geranyl acetate (**7**) and citronellyl acetate (**8**) in trace amounts (< 1%) in the agar medium (Figs 1, 2). In addition to these compounds, tiny amounts of isobutyl alcohol, isoamyl alcohol, isobutyl acetate, ethyl propionate and ethyl acetate were as well found in the SPME analysis.

Three compounds were selected to be tested for field activity (Table 2).

## Field activity

A total of 49,407 *I. typographus* beetles were collected from the three clear-cut areas. A factorial ANOVA output showed for all variables little effect of block and, more importantly, a far from a significant interaction of block and treatment (see supporting information; **S2 Table**). Hence, in the absence of significant interactions, the data from all areas could be pooled for analysis of differences between the treatments, including the control, yielding 48 observations in total (3 blocks × 4 treatments × 4 trap collections). The raw catch was affected by significant deviations from normality and unequal variances, as was also the log and relative catches, but to a lesser degree. The relative catch *r* as the dependent fitted well to the statistical models and gave the largest estimates of power. Use of a generalised form of the linear model for treatment effects (SPSS V28.0 GENLIN with Distribution Tweedie (1,5) and Link Identical) allowed an even better model fit of the treatment effects (Fig 5).

There was a strongly significant overall effect of treatment in the generalised linear model (GzLM) for the relative catch (*r*). Geranyl acetone (**6**) added to the pheromone gave a significantly lower number of *I. typographus* catch according to the GzLM post-hoc test (Fig 5).

Correspondingly, there was a strong effect size estimated by Hedges' *g* value for the reduction due to geranyl acetone (**6**) of −0.96 (95% CI: −1.8 to -0.11). In other words, the catch was one SD lower than that obtained with the pheromone alone, which applied to both the raw data (*x*) and relative catch (*r*). Smaller effects were observed for the other two fungal compounds tested, i.e., sulcatone (**2**) and 2-phenethyl acetate (**5**), which showed no significant difference by the GzLM post-hoc test (Fig 5). Correspondingly, they both had low effect size *g* values < |0.5| (2-phenethyl acetate 0.12 and sulcatone -0.31) with 95% CI including zero, indicating no effect.

## Discussion

### Volatiles produced by *Endoconidiophora rufipennis*

**Isoamyl acetate (1) and sulcatone (2).**    An interesting relationship between the production of isoamyl acetate (**1**) and sulcatone (**2**) was observed. After six days of fungal growth, the production of isoamyl acetate ceased, whereas that of sulcatone increased.

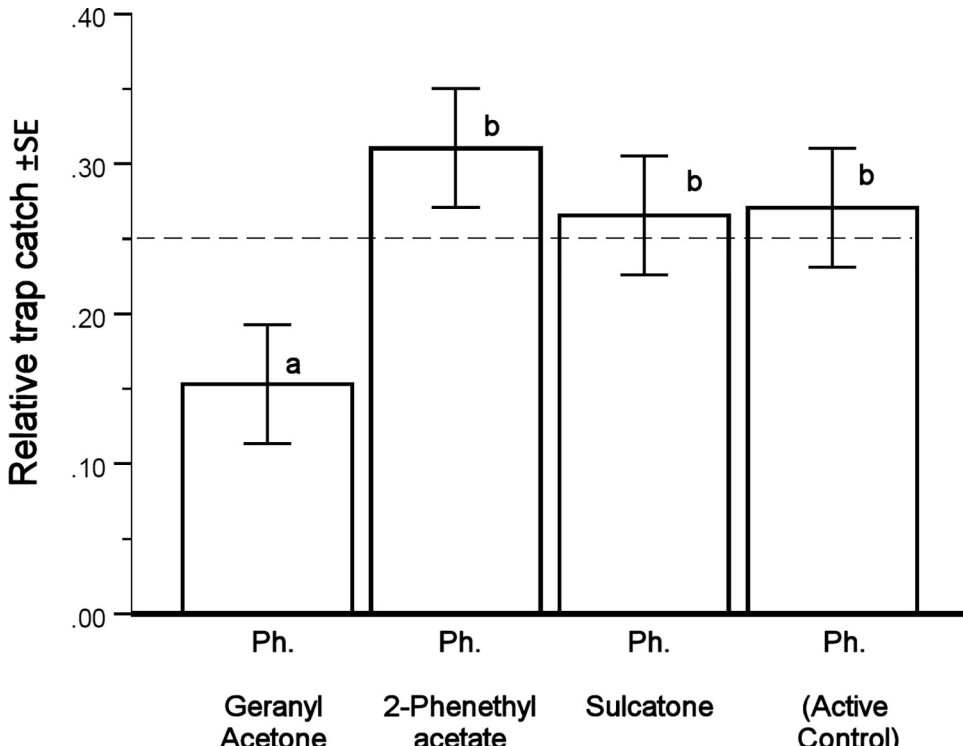

**Fig 5. Mean relative catches (*r*) of *I. typographus*, calculated from EMM (Estimated marginal means) of SPSS UNIANOVA.** The thin dashed reference line at 0.25 shows the zero hypothesis that all four treatments had the same catch (1/4 of total). Only the addition of Geranyl acetone (**6**) to synthetic pheromone (Ipslure®) resulted in a catch significantly lower than the catch of the Control (synthetic pheromone alone). Bars with the same lower-case letters (a or b) did not statistically differ from each other by sequential Sidak post-hoc contrasts following GzLM (Omnibus test [fitted model against the intercept-only model] likelihood ratio $\chi^2$ 10.98, df 3, *p* = 1.2*%, SPSS GENLIN).

Isoamyl acetate has been shown to enhance attraction of the bark beetle *Dendroctonus frontalis* to a mixture of frontalin/*trans*-verbenol/turpentine in the laboratory [32]. Both isoamyl alcohol and isoamyl acetate have been shown to be produced *in vitro* by other blue stain fungus genera, such as *Ophiostoma*, *Endoconidiophora*, *Grosmannia* and *Ceratocystis*, in laboratory experiments [29].

**Sulcatone (2) and sulcatol (3).** Sulcatone (**2**) is a common semiochemical that has been detected from the hindgut in different bark beetles [33]. Sulcatol (**3**) is an aggregation pheromone produced by males of the ambrosia beetle *Gnathotrichus sulcatus* Leconte (Curculionidae), an important economical pest on the Pacific coast of North America [34]. According to the literature, the natural pheromone consists of the sulcatol stereoisomers in the ratio 65:35 *R*/*S*, although the optimal ratio for a bioactive response has been found to be 50:50 [34, 35]. Neither the (*R*)- nor the (*S*)-enantiomer of sulcatol elicit any behaviour on their own: only the mixture is bioactive [36]. This is one of rather few examples where both enantiomers are needed for a synergistic response.

A possible explanation for our findings of both sulcatone (**2**) and sulcatol (**3**) is that sulcatone was reduced to sulcatol by the fungus in a stereospecific way, i.e. (*R*)-sulcatol was formed predominantly. The fact that the enantiomeric proportion of (*S*)-sulcatol first increased during the experiment then declined may be because the fungus acetylated the (*R*)-alcohol to stereoselectively yield (*R*)-sulcatol acetate (**4**) but that this acetylation reaction for some reason abated towards the end of the experiment.

**2-Phenethyl acetate (5).**   This compound has been shown to be produced by several known fungal symbionts of *I. typographus*, such as *E. polonica*, *G. pencillata* and *G. europhioides* [29]. A recent study showed that two classes of odour receptor neurons (ORNs) in *I. typographus* exhibit dose-response activity for 2-phenethyl acetate [28]. 2-Phenethyl acetate (**5**) has been reported to attract *D. frontalis* in laboratory assays when added to non-attractive concentrations of pheromone blends, [32]. Since dose-response activity on ORNs has been reported, it is surprising that no effect was observed in our field tests.

**Geranyl acetone (6).**   Geranyl acetone is a common semiochemical [31] that has been reported as a volatile in 35 plant orders [31]. The bioactivity of geranyl acetone in wood infesting beetles, as attractant or anti-attractant, has been demonstrated [37, 38]. In cerambycid beetles (Longhorn beetles) of subfamilies *Spondylinae* and *Lamiinae*, it has been reported as a field active component of male produced aggregation sex pheromones in more than half a dozen species [39, 40].

Geranyl acetone is strongly EAD (electroantennographic detection)-active towards a congener on larch, i.e., the Asian larch bark beetle *Ips subelongatus* Motschulsky [38]. When field-tested with a three-component pheromone mixture, geranyl acetone significantly reduced the number of *I. subelongatus* captured in traps and was therefore characterised as an anti-attractant [38].

Antennae of *I. typographus* have also been reported to respond to geranyl acetone [29], but electrophysiology activity alone does not infer behavioural activity (attraction, inhibition, or none).

Geranyl acetone was a strong candidate for our field activity test, both owing to its reported anti-attractant effects on *Ips* spp and to being the only compound to peak in its occurrence on the last sampling day. Our results also demonstrated a strong anti-attractant effect of geranyl acetone. This finding is perhaps not surprising but is nevertheless an important result in the search for new and more efficient anti-attractants against *I. typographus*.

**Geranyl acetate (7).**   Geranyl acetone´s oxygenated analogue, geranyl acetate, is a common volatile in various plants [41] and has been reported a semiochemical in Diptera and Hymenoptera [31]. This is the first time this compound has been reported from blue stain fungi.

**Citronellyl acetate (8).**   Citronellyl acetate (**8**) has not been reported as EAD-active to any bark beetle species. However, the synthesis of citronellyl acetate within fungi has been reported before. Earlier research on citronellyl acetate examined different alcoholysis reactions catalysed by lipases from different fungi (*Mucor* sp. and *Rhizopus* sp.) [42, 43].

## Field activity

As we did not have the resources to test all identified fungal volatiles in the field, we selected three compounds based on a combination of scientific rationale and practical reasons. Among the detected volatiles, we had seven compounds to choose from. Isoamyl acetate (**1**) was a strong candidate for the field test but was deselected since it is such a ubiquitous fungal metabolite. Both isoamyl alcohol and isoamyl acetate (**1**) have been shown to be produced by several blue stain fungi, such as *Ophiostoma*, *Endoconidiophora*, *Grosmannia* and *Ceratocystis*, in laboratory experiments. Therefore, these two compounds are not specific for *Endoconidiophora*. We wanted to test biologically active compounds. Thus, geranyl acetone (**6**) and 2-phenethyl acetate (**5**) were selected as they have been reported as electrophysiologically active for *I. typographus* before [28]. Sulcatone (**2**) was selected because it is an isoprene analogue of geranyl acetone (**6**). The three compounds were also selected for practical reasons, i.e., their release rates by the polyethylene dispenser bags were similar and remained steady over the period for

**Table 3. Effect sizes of anti-attractants trapping *Ips typographus* in three trapping studies for the addition of geranyl acetone compared to other host-derived anti-attractants to aggregation pheromone alone (control).**

| Trap bait | Mean catch* | *n* | SD | Effect size (Hedges' *g*) † | Confidence interval *g* | | Study |
|---|---|---|---|---|---|---|---|
| | | | | | Lower | Upper | |
| Pheromone (control) | 3.351 | 6 | 2.530 | – | | | [45] |
| Pher. & 1,8-cineole | 388 | 6 | 219 | −2.4 | −3.5 | −1.3 | |
| Pheromone (control) | 31 | 15 | 21 | – | | | [46] |
| Pher. & verbenone | 4 | 15 | 2 | −1.6 | −2.5 | −0.7 | |
| Pher. & 1,8-cineole | 2 | 15 | 2 | −1.7 | −2.6 | −0.8 | |
| Pheromone (control) | 266 | 19 | 258 | – | | | [47] |
| Pher. & trans-4-thujanol | 40 | 19 | 74 | −1.0 | −1.8 | −0.2 | |
| Pher. & 1,8-cineole | 92 | 19 | 91 | − 0.8 | −1.6 | 0.0 | |
| Pheromone (control) | 1258 | 12 | 896 | – | | | This study |
| Pher. & geranyl acetone | 629 | 12 | 476 | −0.96 | −1.8 | −0.11 | |

*) Absolute catches (*x*).

†) The effect size by Hedges scales the difference between pairs of control and treatment means by division of their pooled SDs [44].

the selected volatiles, enabling fair comparison of the three test compounds (supporting information **S1 Fig**).

The observed activity of geranyl acetone is not only highly significant but represents a new piece of behavioural data for understanding the bark beetle holobiont. For development of scolytid management in forestry practice, the size of the effect needs to be compared to that of other semiochemicals tested with pheromone in traps before extending to larger field tests and testing products. Some recent examples of effect sizes in field tests with *I. typographus* are given in Table 3.

Notably, our new anti-attractant semiochemical geranyl acetone showed an effect size (Hedges' *g*, [44]) close to unity in our trapping field test, similarly to several other old and new anti-attractants, including the host-derived verbenone 1,8-cineole and *trans*-4-thujanol, respectively, already documented for *I. typographus* (Table 3).

As already mentioned in the introduction, the host derived well-known anti-attractant verbenone is an oxygenation product of α-pinene and verbenols giving a clear indication of an old and over-exploited host [17, 31, 46]. It is to be noted that the relative amounts of geranyl acetone are increased five-fold in the last third of the experimental time period. In concurrence with this, the observed field effect could be explained by the evolution of geranyl acetone as yet another signal of unsuitable hosts.

Geranyl acetone is a relatively low-cost compound compared to the known anti-attractants *trans*-conophthorin and thujanol (https://www.sigmaaldrich.com and https://chemtronica.com). The higher volatility of the anti-attractants 1-octen-3-ol, and 1,8-cineole (http://www.chemspider.com/) makes geranyl acetone a more suitable compound for use in dispensers where good longevity is desired.

Anti-attractant volatiles for *I. typographus* have been proven to be active in the field in previous studies. Those anti-attractants were mainly derived from non-host green-leaf and bark volatiles or from old host trees, allowing protection of spruce trees in the field [15, 48–50]. Unfortunately several of the most effective anti-attractants (e.g., *trans*-conophthorin and verbenone), are expensive and are available only in limited amounts from commercial sources (e.g. https://chemtronica.com). Therefore, the discovery of a new and less expensive, yet non-toxic, anti-attractant for controlling outbreaks of *I. typographus* is important.

The next step is to test the dose-response of geranyl acetone (**6**) and to evaluate any synergistic effects together with other earlier known anti-attractants. Similarly to almost all anti-attractant compound combinations tested, strong synergistic effects could improve the efficacy [51, 52]. This would greatly enhance the practical usability and cost-efficiency of using geranyl acetone as a part of an anti-aggregation arsenal for controlling damage caused by *I. typographus*.

## Conclusion

Volatile compounds in ER emissions were analysed over time. Eight of nine identified compounds were isoprenoids. Three of the volatiles identified were subjected to field tests. A strong anti-attractant effect of geranyl acetone (**6**) in overcoming the attractive effect of synthetic aggregation pheromone (Ipslure®) was found, indicating that geranyl acetone may be a strong candidate for future tree protection studies. This is the first study to connect chemical analysis in the laboratory of volatiles from identified fungi of the bark beetle holobiont and insect behaviour in a field experiment.

## Supporting information

**S1 Table.**
(DOCX)

**S2 Table.**
(DOCX)

**S1 File. Experimental procedure.**
(DOCX)

**S1 Fig. Release rate.**
(DOCX)

## Acknowledgments

We thank Dr J-E Englund at SLU, Alnarp for statistical advice and SCA Forest AB for allowing the field trial to be conducted on their land.

## Author Contributions

**Conceptualization:** C. Rikard Unelius.

**Data curation:** Matilda Lindmark, Suresh Ganji, Erika A. Wallin, Fredrik Schlyter.

**Formal analysis:** Matilda Lindmark, Erika A. Wallin, Fredrik Schlyter.

**Funding acquisition:** Erika A. Wallin, C. Rikard Unelius.

**Investigation:** Matilda Lindmark, Suresh Ganji, Erika A. Wallin, C. Rikard Unelius.

**Methodology:** Matilda Lindmark, Suresh Ganji, C. Rikard Unelius.

**Project administration:** C. Rikard Unelius.

**Resources:** C. Rikard Unelius.

**Software:** Fredrik Schlyter.

**Supervision:** Erika A. Wallin, Fredrik Schlyter, C. Rikard Unelius.

**Validation:** Fredrik Schlyter.

**Visualization:** Matilda Lindmark, Fredrik Schlyter.

**Writing – original draft:** Suresh Ganji, C. Rikard Unelius.

**Writing – review & editing:** Matilda Lindmark, Suresh Ganji, Erika A. Wallin, Fredrik Schlyter, C. Rikard Unelius.

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
