## [Decision Letter · Decision Letter 0]

1 Feb 2023

PONE-D-22-20345The title is

Semiochemicals produced by fungal bark beetle symbiont Endoconidiophora rufipennis and the discovery of an anti-attractant for Ips typographusPLOS ONE

Dear Dr. Unelius,

Thank you for submitting your manuscript to PLOS ONE. After careful consideration, we feel that it has merit but does not fully meet PLOS ONE’s publication criteria as it currently stands. Therefore, we invite you to submit a revised version of the manuscript that addresses the points raised during the review process.

Both reviewers found your study to be of considerable interest and worthy of publication subject to minor textual changes. Reviewer 2 has provided quite a long list of these and I draw your particular attention to their detailed comments running between "Lines 204..." to "Lines 477...". Their comments regarding the possible use of generalised linear models and calibration of your SPME data I regard as optional. 

We look forward to receiving your revised manuscript.

Kind regards,

Christopher Walton, Ph.D.

Academic Editor

PLOS ONE

Journal Requirements:

“This work was supported by two grants from The Carl Trygger Foundation, https://www.carltryggersstiftelse.se/this-is-the-carl-trygger-foundation/ and Linnaeus University, Kalmar, Sweden https://lnu.se/en/ (CRU and SG). This research was also financially supported by the Regional Development Fund for Västernorrland and Jämtland/Härjedalen county and the Swedish Agency for Economic and Regional Growth (EAW and ML). FS was supported by project EXTEMIT-K CZ.02.1.01/0.0/0.0/15_003/0000433 financed by OP RDE at the Czech University of Life Sciences, Prague. We thank Dr J-E Englund at SLU, Alnarp for statistical advice and SCA Forest AB for allowing the field trial to be conducted on their land.

“This work was supported by two grants from The Carl Trygger Foundation, (CRU and SG), and Linnaeus University, Kalmar, Sweden. This research was also financially supported by the Regional Development Fund for Västernorrland and Jämtland/Härjedalen county and the Swedish Agency for Economic and Regional Growth (EAW and ML). FS was supported by project EXTEMIT-K CZ.02.1.01/0.0/0.0/15_003/0000433 financed by OP RDE at the Czech University of Life Sciences, Prague. We thank Dr J-E Englund at SLU, Alnarp for statistical advice and SCA Forest AB for allowing the field trial to be conducted on their land.”

“This work was supported by two grants from The Carl Trygger Foundation, https://www.carltryggersstiftelse.se/this-is-the-carl-trygger-foundation/  and Linnaeus University, Kalmar, Sweden https://lnu.se/en/ (CRU and SG). This research was also financially supported by the Regional Development Fund for Västernorrland and Jämtland/Härjedalen county and the Swedish Agency for Economic and Regional Growth (EAW and ML). FS was supported by project EXTEMIT-K CZ.02.1.01/0.0/0.0/15_003/0000433 financed by OP RDE at the Czech University of Life Sciences, Prague. We thank Dr J-E Englund at SLU, Alnarp for statistical advice and SCA Forest AB for allowing the field trial to be conducted on their land.

Reviewers' comments:

Reviewer's Responses to Questions

**Comments to the Author**

1. Is the manuscript technically sound, and do the data support the conclusions?

Reviewer #1: Yes

Reviewer #2: Yes

2. Has the statistical analysis been performed appropriately and rigorously? 

Reviewer #1: Yes

Reviewer #2: Yes

3. Have the authors made all data underlying the findings in their manuscript fully available?

Reviewer #1: Yes

Reviewer #2: Yes

4. Is the manuscript presented in an intelligible fashion and written in standard English?

Reviewer #1: Yes

Reviewer #2: Yes

5. Review Comments to the Author

Reviewer #1: This is an excellent manuscript that describes a very well designed and thoroughly conducted investigation that combines chemical analyses of volatiles from bark beetle fungal symbionts and behavioral responses of those beetles to identified volatiles. A new 'anti-attractant' was identified from a blue stain fungus that may have practical utility against I. typographus beetles. The manuscript is clear and concise. The conclusions are based on the results. The methods can be fully replicated based on the description. The results are clear cut. The data are well placed within the context of the literature.

Couple minor edits:

Line 269: Write out ANOVA when first used.

Line 379: Missing full stop (period).

Reviewer #2: This paper analyzed volatile emissions from laboratory cultures of the blue stain fungus Endoconidiophora rufipennis associated with the North American spruce bark beetle Dendroctonus rufipennis which is closely related to the fungus associated with the Eurasian spruce bark beetle Ips typographus. Several volatile compounds were identified of which 3 were selected and field tested in Europe for attraction or inhibition of Ips typographus. Volatiles of the North American fungus were analyzed as a potential source of new attractants or anti-attractant compounds and because Ips typographus is often intercepted in North America and could eventually be introduced and expand its geographic range there where it may encounter or become associated with this fungal species. One of the fungal volatiles tested significantly inhibited attraction of Ips typographus to its pheromone and thus could have potential as an inexpensive anti-attractant. Overall, the paper is well-written and the methods and results are clearly described and presented. It is the first study to test fungal volatiles’ long range attraction or inhibition in field trapping experiments. Ips typographus is a major pest of spruce trees in Europe and one of the most commonly intercepted bark beetles or woodborers in solid wood packing material and is of great concern as a potential invasive threat to North America. Therefore this study is of interest as it contributes new knowledge about Ips typographus responses to a North American fungus and identifies a compound which may be useful for anti-aggregation. I believe it is of interest and fits the scope of PLOSone and would be suitable for publication with minor revisions. I have no major concerns with the paper, and only a few minor suggestions.

For the statistical analysis of trap catch data, it sounds like the authors tried various transformations and models (raw data, log transformed, relative or proportional catch) and models (all factorial combinations of treatment and block effects) to find a model that fit best. It would seem more appropriate to look at the distribution of the data and use a generalized linear mixed model with the appropriate distribution and link function that fit the observed distribution.

Overall the number of replicates seems somewhat low (only 3 flasks of fungal culture used for volatile collections, and only 3 field sites with one replicate per site and 4 collection periods with treatment positions re-randomized during each collection period for a total of 12 replicates). That being said, a very high number of beetles were captured, there were no time or block by treatment interactions and the sterile media and pure fungal cultures would have little variation, so replication and experimental power appeared to be adequate.

Lines 204-214 - indicate which compounds were provided by which chemical supplier.

Line 222. I suggest adding a brief explanation of how the 3 compounds were selected for field testing in the methods – and not just in the discussion. After “Three chemicals were” insert “selected for testing in the field based on published reports of their activity in bark beetles (see Discussion for detailed rationale). The chemicals were”….

Line 232. Change “For each dispenser, 1 mL of fungal compound was distributed onto the felt in the dispenser bag” to “For each of the test fungal compounds, 1 mL of synthetic pure compound was distributed onto the felt in a dispenser bag”

Line 243. For the Lindgren funnel traps, how many funnel units did they have? What killing agent was used in the collection cups – dry cups with insecticide strip, or wet cups with liquid preservative?

Line 389. What species were odour receptor neurons found in for 2-Phenethyl acetate? Is Ips typographus known to have receptors for this compound?

Line 396. Add a reference for the statement that geranyl acetone has been reported as a common green leaf volatile

Line 398. Add a reference for the statement that geranyl acetate has been demonstrated as an attractant or antiattractant in several taxa.

Line 404. Add a reference for the statement that geranyl acetate as EAD activity in Ips subelongatus.

Line 410-411. Reword sentence – it seems awkward. “but electophsioloical activity alone does not inform of valence of an odour for behavior” Maybe something like “but electophsyiology activity alone does not infer behavioral activity (attraction, inhibition, or none)”

Line 420. Summarize and cite any literature of known physiological or behavioral reponses to geranyl acetate.

Line 440. “have been reported as electophysiologically active” in what species?

Line 467. Add a reference for the statement that Geranyl acetone is relatively low cost compound compared to other anti attractants, and for the anti-attraction activity of 1-octen-3-ol, trans-conophthorin, and thujanol.

Line 468. Add a reference for the lower volatility of geranyl acetone compared to 1,8-cineole

Line 477-478. Add a reference for the limited availabilityia nd high host of other anti-attractants trans-conophthorin and verbenone.

Table 1. In addition to the relative amounts, it might be useful to give the total daily quantity of volatiles emitted. Although there is no quantitative internal standard when using SPME fibers for volatile collection, since the volume of the airspace was the same each day, and the length of time that the SPME fiber was exposed in the airspace, it seems that there should be a way to standardize the total amount of volatiles captured on each day and report them as ng or pg per mL of headspace or something like that? How do the volatile rates compare to the release rates of lures?

Table 2. List the chemical supplier for each compound – perhaps as a footnote

Figure 5. I suggest adding “+ Synth Pher” to each of the treatment labels on the X-axis t clarify and emphasize that each compound was tested in combination with the synthetic pheromone. I also suggest using the letter “a” on the three bars with the highest values and the letter “b” on the bar for Geranyl acetone with the lowest value. The insert with the schematic for the block layout is not necessary and kind of confusing.

6. PLOS authors have the option to publish the peer review history of their article (what does this mean?). If published, this will include your full peer review and any attached files.

Reviewer #1: **Yes: **Lukasz Stelinski

Reviewer #2: No

---

## [Author Response · Author response to Decision Letter 0]

8 Mar 2023

Response to Reviewers

Dear PlosONE, reviewers and Christopher Walton,

Thank you for your valuable comments, we did the following changes in the manuscript: We have now changed the formatting throughout the manuscript to ensure that the manuscript meets PLOS ONE's style requirements, including those for file naming. We have lifted the amended Funding Statement from the manuscript to our cover letter. Thank you for changing the online submission form. Minor edits are now changed throughout the manuscript as follow. Line number is referring to the document called “revised manuscript with track changes”. 

Following minor edits from reviewer #1 has been changed:

Line 269: Write out ANOVA when first used. Changes can be found on Line 278.

Line 379: Missing full stop (period). Changes can be found on Line 422.

Following minor edits from reviewer #2 has been changed:

Lines 204-214 - indicate which compounds were provided by which chemical supplier. Changes can be found on Line 201-209

Line 222. I suggest adding a brief explanation of how the 3 compounds were selected for field testing in the methods – and not just in the discussion. After “Three chemicals were” insert “selected for testing in the field based on published reports of their activity in bark beetles (see Discussion for detailed rationale). The chemicals were”…. Changes can be found on Line 222-224.

Line 232. Change “For each dispenser, 1 mL of fungal compound was distributed onto the felt in the dispenser bag” to “For each of the test fungal compounds, 1 mL of synthetic pure compound was distributed onto the felt in a dispenser bag” Changes can be found on Line 218-220.

Line 243. For the Lindgren funnel traps, how many funnel units did they have? What killing agent was used in the collection cups – dry cups with insecticide strip, or wet cups with liquid preservative? 

Clarification can be found on Line 251-260, no killing agent was utilized in the field experiment. But we are aware that this may affect the results (usually lower catches due to that insects may escape or predadors may enter and the trap and kill/eat the caught spruce bark beetles).

Line 389. What species were odour receptor neurons found in for 2-Phenethyl acetate? Is Ips typographus known to have receptors for this compound? Clarification can be found on Line 431. 

Line 396. Add a reference for the statement that geranyl acetone has been reported as a common green leaf volatile Clarification can be found on Line 439.

Line 398. Add a reference for the statement that geranyl acetate has been demonstrated as an attractant or antiattractant in several taxa. Assuming that the reviewer mean geranyl actone clarification can be found on Line 441-444 and refs added in Line 444. 

Line 404. Add a reference for the statement that geranyl acetate as EAD activity in Ips subelongatus. Assuming that the reviewer mean geranyl actone as he/she refer to line 404 in the geranyl acetone paragraph. Reference added on Line 447.

Line 410-411. Reword sentence – it seems awkward. “but electophsioloical activity alone does not 2-alone does not infer behavioral activity (attraction, inhibition, or none)” Changes can be found on Line 451-3.

Line 420. Summarize and cite any literature of known physiological or behavioral reponses to geranyl acetate. Reference inserted on Line 462-3, geranyl acetate is a very common floral volatile and a very common semiochemical in diptera and Hymenoptera.

Line 440. “have been reported as electophysiologically active” in what species? Clarification can be found on Line 482.

Line 467. Add a reference for the statement that Geranyl acetone is relatively low cost compound compared to other anti attractants, and for the anti-attraction activity of 1-octen-3-ol, trans-conophthorin, and thujanol. Clarification can be found on Line 522-527. The price for thujanol is 250 USD for 5 gram and tr-conophthorin can only be bought from a small company 

https://chemtronica.com. ).

Line 468. Add a reference for the lower volatility of geranyl acetone compared to 1,8-cineole. Reference can be found on Line 522-527.

Line 477-478. Add a reference for the limited availability and high cost of other anti-attractants trans-conophthorin and verbenone. 

Clarification can be found on Line 532-536 trans-conophthorin can only be bought from a small company (https://chemtronica.com. ). and verbenone is used a lot in US for the mountain pine beetle so the supply of this natural product is depleted everywhere. 

Table 1. In addition to the relative amounts, it might be useful to give the total daily quantity of volatiles emitted. Although there is no quantitative internal standard when using SPME fibers for volatile collection, since the volume of the airspace was the same each day, and the length of time that the SPME fiber was exposed in the airspace, it seems that there should be a way to standardize the total amount of volatiles captured on each day and report them as ng or pg per mL of headspace or something like that? How do the volatile rates compare to the release rates of lures? 

The purpose of Table 1 is to indicate how the relative emitted proportions of the compounds change over time. Nothing else, so no change.

Table 2. List the chemical supplier for each compound – perhaps as a footnote 

Not done, we believe it is enough to give the specified suppliers in the M&M section 

Figure 5. I suggest adding “+ Synth Pher” to each of the treatment labels on the X-axis to clarify and emphasize that each compound was tested in combination with the synthetic pheromone. I also suggest using the letter “a” on the three bars with the highest values and the letter “b” on the bar for Geranyl acetone with the lowest value. The insert with the schematic for the block layout is not necessary and kind of confusing. 

Removing an inset too small is a good suggestion, thanks!

Logically, the different treatment odours are added (“+”) to the control (pheromone), not the other way around. Adding 4 times the present longer acronym to the x-axis labels adds visual noise (overcrowding). Thus, a shorter acronym is used “Ph.” (synthetic is not vital as only 1 type of pheromone is used).

“a” is normally used before “b” from the alphabetic order inherited from the Phoenicians.

The changed fig. 5 can be found on Line 368-69

---

## [Decision Letter · Decision Letter 1]

20 Mar 2023

Semiochemicals produced by fungal bark beetle symbiont Endoconidiophora rufipennis and the discovery of an anti-attractant for Ips typographus

PONE-D-22-20345R1

Dear Dr. Unelius,

We’re pleased to inform you that your manuscript has been judged scientifically suitable for publication and will be formally accepted for publication once it meets all outstanding technical requirements.

Kind regards,

Christopher Walton, Ph.D.

Academic Editor

PLOS ONE

Additional Editor Comments (optional):

Reviewers' comments:

Reviewer's Responses to Questions

**Comments to the Author**

1. If the authors have adequately addressed your comments raised in a previous round of review and you feel that this manuscript is now acceptable for publication, you may indicate that here to bypass the “Comments to the Author” section, enter your conflict of interest statement in the “Confidential to Editor” section, and submit your "Accept" recommendation.

Reviewer #2: All comments have been addressed

2. Is the manuscript technically sound, and do the data support the conclusions?

Reviewer #2: Yes

3. Has the statistical analysis been performed appropriately and rigorously? 

Reviewer #2: Yes

4. Have the authors made all data underlying the findings in their manuscript fully available?

Reviewer #2: Yes

5. Is the manuscript presented in an intelligible fashion and written in standard English?

Reviewer #2: Yes

6. Review Comments to the Author

Reviewer #2: (No Response)

7. PLOS authors have the option to publish the peer review history of their article (what does this mean?). If published, this will include your full peer review and any attached files.

Reviewer #2: **Yes: **Therese Poland

---

## [Editor Report · Acceptance letter]

27 Mar 2023

PONE-D-22-20345R1 

Semiochemicals produced by fungal bark beetle symbiont *Endoconidiophora rufipennis* and the discovery of an anti-attractant for *Ips typographus*

Dear Dr. Unelius:

I'm pleased to inform you that your manuscript has been deemed suitable for publication in PLOS ONE. Congratulations! Your manuscript is now with our production department. 

Kind regards, 

on behalf of

Dr. Christopher Walton 

Academic Editor

PLOS ONE